# Antibacterial Properties of *Eucalyptus globulus* Essential Oil against MRSA: A Systematic Review

**DOI:** 10.3390/antibiotics12030474

**Published:** 2023-02-27

**Authors:** Shakthi Elangovan, Poonam Mudgil

**Affiliations:** School of Medicine, Western Sydney University, Campbelltown, NSW 2560, Australia

**Keywords:** methicillin-resistant *Staphylococcus aureus*, MRSA, *Staphylococcus aureus*, antibiotics, antimicrobial resistance, eucalyptus oil, eucalyptus essential oil, *E. globulus* essential oil

## Abstract

Antimicrobial resistance is a huge threat against the public health sphere and is a major cause of global mortality and morbidity. Antibiotic misuse and overuse have led to the development of many resistant bacterial strains. One particular bacterium of concern is methicillin-resistant *Staphylococcus aureus* (MRSA), which is the most common resistant bacteria in humans. Antibiotic development has been unable to keep up with the rapid evolution of antibiotic-resistant organisms, and there is an urgent need to identify alternative agents to combat this problem. The purpose of this systematic review is to explore the literature on the antibacterial properties of *Eucalyptus globulus* essential oil against MRSA. The articles used in this review were obtained through a systematic search of the literature across four databases, with the timeline being between 2002 and 2022. Twenty studies were included in this review, which used various methods to investigate the antibacterial properties of *E. globulus* essential oil, alone or in combination with other agents, against MRSA. The findings suggest that *E. globulus* essential oil has antibacterial properties against MRSA, which can be enhanced when used in combination with other agents, such as other essential oils and antibiotics.

## 1. Introduction

Antibiotic resistance is a phenomenon in which bacteria are able to evade antibiotics via various mechanisms, including neutralisation of the drug, excreting the drug, modifying structural components to prevent the drug from acting on the cell and DNA transfer between bacteria [1,2]. Drivers of this resistance include overuse and misuse of antibiotics in clinical and agricultural settings [1,3]. When antibiotics are used, they act on drug-sensitive bacteria, eradicating them and leaving behind resistant bacteria, which reproduce and proliferate [3]. Since the advent of modern antibiotics in the 1940s, bacteria have developed resistance to almost all available antibiotics and pose a huge threat to public health [3]. It is a major cause of global mortality and is classified by the World Health Organisation (WHO) as being in the top ten threats to global health [4].

Methicillin-resistant *Staphylococcus aureus* (MRSA) was first described in 1961 and, since then, it has become the most common resistant strain of bacteria in healthcare [2,5]. MRSA causes a wide range of infections, with the most common sites being the skin and subcutaneous tissue, followed by sites, including, but not limited to, the meninges, endocardium and bone [5,6].

The first three decades of MRSA cases were predominantly in those who had contact with hospitals, but in the 1990s, infections began to emerge in community settings in those that had no healthcare contact, presenting a major problem to the public health sphere. Management of MRSA depends on the specific disease and involves strict infection prevention methods and antibiotic administration. In the hospital setting, intravenous vancomycin is often the drug of choice, with daptomycin being a suitable alternative. However, cases of vancomycin-resistant *Staphylococcus aureus* have also emerged, complicating the situation even further [5].

The discovery of new antibiotics has not been able to keep up with the emergence of these resistant strains, highlighting the need to urgently find alternatives to antibiotics [7]. One such alternative is essential oils. Essential oils are complex mixtures, generally composed of over 20 different compounds. Over 3000 essential oils have been identified and each is composed of varying compounds and quantities. They have been used for centuries in traditional medicine to treat various conditions, from cuts and wounds to coughs and colds. Many essential oils possess antibacterial qualities that can be, in part, attributed to the low molecular weight of their active compounds and their lipophilic nature, enabling them to cross cell membranes and have cytotoxic effects [8]. The mechanism of action of essential oils differs from the mechanism of antibiotics, in that it inhibits various physiological and structural components rather than a single target like antibiotics. Examples of targets include inhibition of cell membranes, efflux pumps, biofilm and motility. The combination of various mechanisms complements one another, leading to a greater inhibition of bacterial growth compared to traditional antibiotics. This highlights their potential as alternative agents to antibiotics [9]. Additionally, evidence exists to demonstrate that antibacterial properties of essential oils can be enhanced when they are used in combination with other agents, such as essential oils and antibiotics [10].

*Eucalyptus globulus* is a plant that belongs to the *Myrtaceae* family. The essential oil derived from this plant is used widely around the world for many purposes, such as for pharmaceuticals, perfumes, food products and cosmetics. The oil has been shown to possess antibacterial, anti-inflammatory and antioxidant properties [11]. It has antibacterial effects against a broad range of microorganisms and, though the exact mechanism of action is unknown, multiple factors have been proposed, such as the ability to disrupt cell wall and membranes, leading to ATP and metabolite leakage. Additionally, the hydrophobic nature of the oil enables increased cell permeability, leading to bacterial cell leakage [12]. However, there remains a research gap, wherein it is unclear how effective these oils are against multidrug-resistant bacteria, in particular against MRSA. Thus, the aim of this systematic review was to identify whether eucalyptus essential oil, alone or in combination with other compounds, shows antimicrobial effects against MRSA.

## 2. Results

### 2.1. Search Results

Searching the four databases using the search strategy yielded a total of 242 articles. After duplicates were removed, 113 articles remained. The titles and abstracts of these articles were screened to yield 45 results. After reading the full texts of these 45 results, 25 articles were removed, due to reasons, such as not testing MRSA, not using *E. globulus* species, not using the essential oil and not being a primary article; 20 articles were included in this review, as these directly answered the focused question. The study selection process is presented in Figure 1 and the data extracted from the 20 included studies are presented in Table 1.

### 2.2. Effectiveness of Essential Oil against MRSA Alone as Per Different Methodology

#### 2.2.1. Chemical Composition—GC/MS

Six studies studied the chemical composition of the oil using gas chromatography/mass spectrometry (GC/MS) [13,19,22,25,26,30] and one study used gas chromatography alone [28]. The composition of the oils varied between studies, between oils extracted from different parts of the plant and between various plant stages. All investigations, except for the fruit oil tested by Mulyaningsih et al., 2011 [25], found that 1,8 cineole was the main component of the extracts, with percentage values ranging from 32.19% [28] to 86.51% [25]. The values for 1,8 cineole were 84.8%, as per Acs et al., 2016 [13], 81.93%, as per Hamoud et al., 2012 [19], 84.2%, as per Horvath et al., 2011 [22], 86.51%, as per Mulyaningsih et al., 2010 [25], 32.19%, as per Salem et. al. 2018 [28] and 47.2%, as per Tohidpour et al., 2010 [30].

Salem et al., 2018 [28] and Tohidpour et al., 2010 [30] extracted the oils from “aerial parts” and did not specify that the extracts were from the leaf. Mulyaningsih et al., 2011 was the only study that tested oil from both the leaves and the fruit of *E. globulus* [25]. In the fruit oil, they identified aromadendrene as the main component, with 31.17%, followed by 1,8 cineole at 14.55%, whereas the two main components of the leaf oil were 1,8-cineole (86.51%), followed by α-pinene (4.74%).

Salem et al., 2018 was the only study to investigate the composition of oil based on the plant stage [28]. They discovered that the chemical composition differed depending on the plant stage. 1,8-cineole was the major compound at vegetative and full flowering (32.19%) but *p*- cymene was the major compound at fructification stages (37.82%).

#### 2.2.2. MIC

Of the 19 studies, 13 determined the MIC of eucalyptus oil, with a large range of values from 0.032 mg/mL [7] up to 307 mg/mL [23]. Four studies carried out the procedure as per CLSI guidelines [17,21,23,26], one study used the microdilution method as per German DIN regulation [19], one study used the microdilution method as per De Lima Marques et al. (2015) [28], one study used the agar dilution method approved by NCCLS [30] and all other studies did not specify. All 13 studies concluded that eucalyptus essential oil has antibacterial properties against MRSA. The MIC values for studies that only tested the oil were 5.6 mg/mL as per Acs et al., 2016 [13], 0.33 mg/mL as per Ali et al., 2022 [14], 200 mg/mL as per Bouras et al., 2016 [15], 0.313 *v*/*v*% as per Cui et al., 2021 [17], 10 mg/mL as per Hamoud et al., 2012 [19], 32 µg/mL as per Iseppi et al., 2021 [7] and 85.6 µg/mL as per Tohidpour et al., 2010 [30].

Kwiatkowski et al., 2019 [23], Simsek et al., 2017 [29], Hendry et al., 2009 [21] and Merghni et al., 2018 [24], tested 1,8-cineole. Kwiatkowski et al., 2019 revealed the MIC was 307.00 mg/mL against Mupirocin-sensitive MRSA and 57.56 mg/mL against Mupirocin-resistant MRSA [23]. Simsek and Duman 2017 demonstrated that the MIC of 1,8 cineole was 128 g/L [29]. 

Merghni et al., 2018 [24] and Hendry et al., 2009 [21] tested both the MIC of the oil and its main component 1,8 cineole. Merghni et al., 2018 demonstrated that *E. globulus* oil had an MIC of 10 mg/mL and 1,8-cineole 1.25 mg/mL [24], whilst Hendry et al., 2009 showed that the MIC of *E. globulus* oil was 2 g/L and 1,8 cineole was 64 g/L [21]. 

Mulyaningsih et al., 2010 demonstrated that the eucalyptus leaf oil had an MIC between 2000 and >4000 µg/mL, fruit oil had an MIC of 250 µg/mL and aromadendrene was 0.25–1 mg/mL [26]. Salem et al., 2018 tested the oil at various growing stages to reveal MIC at vegetative stage 2 mg/mL, full flowering 4 mg/mL and fructification 4 mg/mL [28].

#### 2.2.3. Biofilm 

Five studies examined the ability of eucalyptus oil to reduce MRSA biofilm formation [7,20,21,24,27].

Hendry et al., 2012 examined wipes containing 5% and 2% eucalyptus oil, 2% chlorhexidine digluconate (CHG) and 70% isopropyl alcohol (IPA) [20]. They revealed that microbial biofilms were eliminated within 10 min (*p* < 0.05) when exposed to 2% EO formulation and within 5 min for 2% EO compared with the control, which took greater than 30 min. Hendry et al. (2009) revealed that eucalyptus oil was more effective at reducing biofilm, as the MIC of eucalyptus oil required to remove the biofilm was lower than that of 1,8 cineole [21]. 

Iseppi et al., 2021 tested eucalyptus oil alone to reveal it had an optical density of 0.1 at 570 nm, and this was even more effective in combination with *Melaleuca alternifolia* (tea tree) essential oil, in which the optical density was <0.05 (*p* < 0.001) [7]. Merghni et al., 2018 measured the percentage reduction in eucalyptus oil and its main component, 1,8 cineole [24]. 1,8 cineole was more effective than the whole oil, with a mean percentage reduction in the oil of 74.74% to 89.15% compared with 1,8 cineol, which was 77.46% to 90.81%. Punitha et al., 2014 also demonstrated that biofilm was inhibited considerably by eucalyptus oil, with average zones of inhibition ranging from 12.2 mm to 26.2 mm [27].

#### 2.2.4. Zones of Inhibition 

Three methods were used to study the zones of inhibitions: well diffusion, direct bioautography assays and disc diffusion, which was used by the majority. 

Nine studies used the agar disc diffusion method to determine the zone of inhibition of *E. globulus* essential oil [7,15,16,18,24,27,28,30,31], with all except one [16] revealing antibacterial activity. Bouras et al., 2016 revealed a diameter between 8 and 14 mm, which, according to their classification, indicated that it was very sensitive (++) for all MRSA strains tested [15]. Chao et al., 2008 revealed that the zone of inhibition for eucalyptus oil was 0 mm; this was the only study that did not produce any activity on disc diffusion [16]. Farsi et al. demonstrated that the eucalyptus oil inhibition zone’s diameter was 11.33 mm and was significantly increased (*p* < 0.001) when loaded onto silica dioxide nanoparticles, with a diameter of 18.66 mm [18]. Iseppi et al., 2021 revealed that the inhibitory zone diameter was between 11 and 20 mm [7]. Merghni et al., 2018 investigated the inhibition zones of both eucalyptus oil and its main component 1,8 cineole to reveal that the oil had a reduced zone of inhibition compared to 1,8 cineole [24]. Eucalyptus oil diameter ranged from 10.7 mm to 26.3 mm, whilst 1,8-cineole was ≥29 mm. Salem et al. demonstrated varying levels of antibacterial activity between the oil, depending on the plant stage [28]. Inhibition zones reached up to 38 mm during vegetative stage but only 24 mm during flowering and fruitification. Punitha et al., 2014 examined the zone of inhibition in relation to biofilm and demonstrated that average zones of inhibition ranged from 12.2 mm to 26.2 mm [27]. Tohidpour et al., 2010 revealed that the inhibition zone of *E. globulus* oil was 8 mm [30], and for Warnke et al., it was 14 mm [31].

Ali et al., 2022 used the well diffusion method to determine zones of inhibition for multiple essential oils, including *Syzygium aromaticum, Eucalyptus globulus, Cinnamomum verum* and *Ferula assafoetida*, to reveal that eucalyptus oil recorded the second-highest zone of inhibition of 18.67 ± 2.51 mm [14].

Horvath et al. trialled various volumes of the essential oil using direct bioautography assays to determine the inhibition zones and revealed that eucalyptus oil produced 0 mm for 1 μL oil, 2.5 mm for 5 μL oil and 6.5 mm for 10 μL oil [22].

#### 2.2.5. Vapour Phase 

Only one study, Acs et al., 2016, showed the effects of *E. globulus* oil in the vapour phase [13]. They demonstrated that eucalyptus oil did not present inhibition against any test bacteria, even in a 1500 μL/L concentration, which differed from their tube dilution results, which showed antibacterial effects against MRSA in liquid medium.

#### 2.2.6. Time-Kill Assays 

Three studies [7,17,19] used time–kill assays. 

Iseppi et al., 2021 [7] showed synergistic activity between tea tree oil and eucalyptus oil, with bacterial load reduction obtained at low concentrations in both synthetic and natural compounds. The optical density at 595 nm was 0.2 after 24 h and was not statistically significant for eucalyptus oil alone or for oxacillin alone; however, it was for the combination mentioned above. The optical density of oxacillin was 0.8 after 24 h. The optical density for tea tree oil in combination with eucalyptus oil was almost >0.05 after 24 h.

Hamoud et al., 2012 [19] demonstrated that Olbas oil exhibited a bactericidal effect against MRSA (reduction of 3 × log10 cfu/mL) at a concentration of 8 × MIC (10 mg/mL Olbas) after 24 h. Lower concentrations only exhibited a weak bacteriostatic effect within the first six hours, which was then followed by significant regrowth.

Cui et al., 2021 [17] did not trial eucalyptus oil for their time–kill assays.

### 2.3. Effectiveness of E. globulus Essential Oil in Combination with Other Agents

Seven studies [7,17,18,19,20,23,26] examined the effects of eucalyptus oil when used in combination with other compounds. Hamoud et al., 2012 [19] and Iseppi et al., 2021 [7] examined the synergy between eucalyptus essential oil and various other essential oils. Hamoud et al., 2012 examined Olbas, an essential oil distillate composed of peppermint oil, eucalyptus oil, cajuput oil, juniper berry oil and wintergreen oil, which had greater antibacterial effects than eucalyptus alone [19]. Iseppi et al., 2021 demonstrated synergistic activity between *Melaleuca alternifolia* (tea tree) essential oil and eucalyptus essential oil [7]. 

Cui et al., 2021 [17], Iseppi et al., 2021 [7] and Kwiatkowski et al., 2019 [23] studied the synergistic effects of eucalyptus oil and antibiotics, with all studies concluding a greater antibacterial effect than with each agent acting alone. Cui et al., 2021 revealed that there was a low level of synergy between eucalyptus oil and antibiotics, including vancomycin, streptomycin, gentamicin and tetracycline [17]. Iseppi et al., 2021 revealed synergy between eucalyptus oil and oxacillin [7], and Kwiatkowski et al., 2019 demonstrated synergy between mupirocin and 1,8-cineole [23]. 

Mulyaningsih et al., 2010 examined the combination of eucalyptus essential oil components, noting synergy between the two major components, aromadendrene and 1,8 cineole [26]. Farsi and Alaidaroos 2022 examined eucalyptus oil loaded in silica dioxide nanoparticles, which revealed a significant increase in the diameters of inhibition zones (*p* < 0.001) compared with the diameters of the individual agents [18]. Hendry et al., 2009 demonstrated synergistic activity between CHG and both eucalyptus oil and 1,8-cineole against suspensions of MRSA [21].

## 3. Discussion

Antibiotic resistance has become a major cause of global mortality and is classified by the World Health Organisation as being in the top ten threats to global health (WHO, 2020) [4]. It is a process wherein bacteria evade antibiotics via various mechanisms, the drivers of which include overuse and abuse of antibiotics in clinical and agricultural settings [1,3]. Since the 1940s, with the advent of modern antibiotics, bacteria have developed resistance to almost all available antibiotics and, thus, there is an urgent need to identify alternative compounds to combat this issue. An example of such an alternative is essential oils. Eucalyptus essential oil is composed of a myriad of volatile compounds, each of which have varying levels of antibacterial activity. The oil in its whole form has been proven to possess antibacterial qualities against MRSA and its biofilm. Its antibacterial properties can be enhanced when used in conjunction with other agents, such as other essential oils and antibiotics.

### 3.1. Composition of E. globulus Essential Oil

Essential oils are composed of complex combinations of volatile plant compounds, with the main compounds often being terpenoid and phenylpropanoid derivatives [32]. The chemical composition of the oils can be determined using gas chromatography along with other analytical tools, such as mass spectrometry [32]. Based on the results of the gas chromatography/mass spectrometry, the composition of the eucalyptus essential oils varies greatly depending on which part of the plant the oil was extracted from. For example, Mulyaningsih et al., 2011 reveal that the main component of the oil extracted from the leaves of the plant is 1,8 cineole, whereas the main component of oil extracted from the fruit is aromadendrene and only contains one-fifth the amount of 1,8 cineole as the leaf oil [25].

The composition also varies depending upon the growing stage of the plant; for example, Salem et al. showed that 1,8-cineole was the major compound at vegetative and full flowering, but *p*-cymene was the major compound at fructification stages [28]. Thus, this may explain the reason for such varied values among the studies that did not specify where the oil was extracted from. Salem et al., 2018 [28] and Tohidpour et al., 2010 [30] had significantly lower values and did not specify where from the plant they retrieved the oil, only mentioning “aerial parts”, which includes any part of the plant above the ground, such as fruits, leaves, stems and flowers, all of which have vastly different compositions [33]. This is also suggested by the results of Mulyaningsih et al.’s 2011 study of both leaves and fruit oils [25]. Additionally, other than Salem et al., none of the studies indicated what growing stage the plants were in, and this is also likely to contribute to the differences in composition, as indicated by the results of Salem et al. [28].

Thus, the plant growing stage and the part of the plant the oil is extracted from should be specified to accurately correlate antibacterial activity with individual oil constituents to ensure accuracy and reproducibility of the results [28]. The method of extraction may also change the composition and, thus, this should also be noted [32].

### 3.2. MIC of E. globulus Essential Oil against MRSA Strains

The minimum inhibitory concentration (MIC) is the lowest concentration of an antimicrobial agent that is required to visibly inhibit the growth of a bacterium after incubation overnight [34]. The MIC values of the whole essential oil range from 0.032 mg/mL [7] to 200 mg/mL [15], being 300-times the value of the second-highest MIC, which was only 10 mg/mL [15]. The values for all other studies were relatively similar, within a range of 0.032 mg/mL to 10 mg/mL [7,13,14,15,17,19,30].

The studies that only tested 1,8 cineole had much higher MIC values than those of the whole oil [21,23,29]. The lowest MIC for 1,8 cineole was 1800-times the value of the lowest MIC for the whole oil. This was 230-times higher than the MIC of aromadendrene, suggesting that the antibacterial activity of 1,8 cineole is much lower than that of the whole oil and aromadendrene. Additionally, the MIC of eucalyptus leaf oil was 16-times higher than that of fruit oil. As demonstrated by Mulyaningsih et al., 2011, this may be explained by the fact that the main component of the leaf oil is 1,8 cineole, whereas the main component of the fruit oil is aromadendrene [25]. Aromadendrene has more antibacterial potency than 1,8 cineole, resulting in a lower MIC of the leaf oil [25].

The MIC of eucalyptus oil varies during different phases of the plant growth cycle, with MIC during the vegetative stage being half than that of full flowering and fruitification [28]. This may also be explained by the changes in composition during these phases, as explained in the previous paragraph. Additionally, variations in MIC may be explained by factors, including, but not limited to, the MRSA strain it was tested against, the geographic origin of the plant, the period in which it was harvested, how it was processed and the conditions in which it was stored [35]. All these factors have an impact on the composition of the oils as well as the proportion and interaction of their volatile molecules [35,36].

### 3.3. Zones of Inhibition Produced by E. globulus Essential Oil

The disc diffusion test is used to determine the susceptibility of a microorganism to an antimicrobial agent [37]. The diameter of growth that was inhibited is known as the zone of inhibition [37]. The zone of inhibition diameters for eucalyptus essential oil vary between studies, with values ranging from 0 mm to 38 mm. Only one study [16] showed that the oil did not produce any inhibition, while another study [22] demonstrated no inhibition for small volumes (1 μL) of oil but inhibition for greater volumes. Since 30 μL of oil was used in the first study [16], insufficient volume of the oil cannot be the reason for non-inhibition. It is notable that the disc diffusion method is a qualitative method that is not very effective for essential oils because their volatile nature reduces their potency, and, thus, the results may show reduced antibacterial activity [38].

### 3.4. Effectiveness of Combination of Agents against MRSA Strains

Synergy is achieved when the sum of the antibacterial activity of two agents is greater than when acting alone [39,40]. This may be attributed to the presence of multiple active antibacterial constituents and interactions between various constituents of both oils [39,40], the combination of which may have various actions, such as increasing the solubility and/or bioavailability of one or more constituents of the oil or acting on different targets, leading to an enhanced antibacterial effect [39]. Synergy is noted between various agents, such as Olbas (composed of peppermint oil, eucalyptus oil, cajuput oil, juniper berry oil and wintergreen oil) [19], between tea tree oil and eucalyptus oil [7], various antibiotics and eucalyptus oil [7,17,23,41].

Synergy is also noted between eucalyptus oil loaded in silica dioxide nanoparticles [18]. The encapsulation of the essential oils reduces their volatility by decreasing their sensitivity to moisture, oxygen, light and heat and, thus, can enhance their antibacterial effects [40]. Additionally, eucalyptus oil tested with 2% chlorhexidine digluconate (CHG) and 70% isopropyl alcohol (IPA) within a wipe showed synergy between eucalyptus oil and CHG against MRSA grown in biofilm and planktonic cultures [20]. The wipes that contained eucalyptus oil, CGH and IPA were significantly quicker and more effective at eliminating biofilm than the wipes that contained only CHG and IPA [20]. This shows that eucalyptus essential oil has potential to enhance the efficacy of hard surface disinfectant wipes that can be used in clinical settings to minimise the risk of hospital-acquired infections. 

### 3.5. Biofilm Inhibition by E. globulus Essential Oil

Biofilm enables bacteria to evade antibiotics and the host immune system and is, thus, a cause of nosocomial infections [42]. It enables the bacterium to embed itself into the biofilm and establish planktonic forms [42]. Hendry et al. (2012) [20] is the only study that tested *E.globulus* oil against planktonic growth and demonstrated that it is effective against both biofilm and planktonic modes of growth [21]. They also revealed that the MIC of eucalyptus oil required to remove the biofilm was lower than that of 1,8 cineole, in contrast with Merghni et al., 2018, who revealed that the opposite is true, with 1,8 cineole having a greater mean percentage reduction in biofilm and, thus, being more effective than the whole oil [24]. Merghni et al., 2018 [24] did not measure the MIC and, thus, a direct comparison with Hendry et al., 2012 [20] may not be accurate to make a conclusion about which agent is more effective at biofilm inhibition. Punitha et al., 2014 also demonstrated that biofilm was inhibited considerably by eucalyptus oil [27]; however, as they measured the effectiveness using zones of inhibition, it is once again inaccurate to compare these results with those of the other studies. Additionally, a combination of eucalyptus oil and tea tree oil is more effective at reducing biofilm than either agent alone [7].

### 3.6. Vapour Phase of E. globulus Essential Oil

The vapours from essential oils have shown potential to possess antibacterial properties, which can be altered in the liquid phase; however, only a limited number of studies have tested the oils in their vapour phase [43,44]. Of the studies included in this review, Acs et al., 2016 was the only one that tested the antibacterial properties of eucalyptus oil in the vapour phase, the rationale for which was to evaluate it as a potential treatment for respiratory tract infection, wherein the vapour can be inhaled [13]. Eucalyptus oil did not inhibit any bacterial growth, even at high concentrations, which contradicted the results of the tube dilution method, indicating that it may be more potent in liquid form rather than vapour form. The vapour form places emphasis on the testing of volatile compounds rather than all compounds within the oil and, thus, these results may demonstrate that the concertation and potency of the volatile compounds within the oil have less antibacterial properties than the other constituents of the oil [13].

### 3.7. Time–Kill Test to Determine Bactericidal Potential of E. globulus

The time–kill test is a robust test for determining the bactericidal potential of an agent [45]; however, only two studies utilised this method to test eucalyptus oil [7,19]. Hamoud et al., 2012 demonstrates that there is a dose-dependent antibacterial effect against MRSA [19]. Iseppi et al. did not test the oil at various concentrations, but rather revealed that there is a time-dependent antimicrobial effect [7].

### 3.8. Strengths and Limitations of Study

This study has many strengths. Many of the included studies follow laboratory guidelines, such as CLSI, thus ensuring high quality. Additionally, the studies tested various parameters, such as variations between different parts of the eucalyptus plant, oils from different plant growth stages, biofilm inhibition, antibacterial properties of the oil in vapour phase and combinations of agents. This provides a more holistic understanding of the antimicrobial properties of the oil and allows for a clearer understanding of its multifaceted nature. These studies were carried out in different parts of the world, ensuring that antibacterial effects of eucalyptus oil are ubiquitously reported.

This study also has some limitations. Only four databases were searched and, thus, some articles may have been missed. Additionally, only English texts were included, and this may have led to exclusion of other relevant studies. Furthermore, there is limited research on this topic and, thus, the number of studies included in this review is minimal. This diminishes the ability to make conclusive judgments and, thus, more data are required. Within these studies, there are a wide variety of methods used, which, although providing valuable insight into various paraments, affects the overall reliability and applicability of the data.

## 4. Materials and Methods

### 4.1. Focused Question

Does eucalyptus essential oil, alone or in combination with other compounds, exhibit antibacterial effects against MRSA?

### 4.2. PICO Question

Population—MRSA strains

Intervention—eucalyptus essential oil

Control—no oil/antibiotic

Outcome—antimicrobial effects

### 4.3. Search Strategy

The literature search was performed on 30 June 2022 and included four databases: PubMed, Web of Science, MEDLINE and Embase. The timeline restricted to twenty years, from 1 July 2002 until 30 June 2022. Keywords included “(Antimicrobial OR antibacterial OR antifungal OR antiviral) AND (eucalyptus oil OR eucalyptus OR aromadendrene OR 1,8 cineole OR globulol) AND (MRSA OR methicillin resistant staph aureus OR methicillin resistant staphylococcus aureus OR methicillin resistant s aureus)”.

### 4.4. Eligibility Criteria

#### 4.4.1. Inclusion Criteria

Only full-text primary articles in English were included. All methodologies, such as vapour and broth microdilution, were included. Studies that used the oil as a whole or isolated compounds from the oil were included. Studies that used oil as the only agent or used combination of oil with other agents and various forms of oil, e.g., fruit oil and leaf oil, were all included.

#### 4.4.2. Exclusion Criteria

Grey literature, conference abstracts, posters and review articles were excluded. Studies that used eucalyptus leaf extracts, rather than essential oil, were excluded, as were studies that tested species of *Eucalyptus* other than *E. globulus* or tested the agents against pathogens other than MRSA.

### 4.5. Study Selection

Titles and abstracts were reviewed and inclusion and exclusion criteria were applied to ensure only relevant articles were included, after which full text was screened to further exclude any articles that did not align with the criteria. The search selection was carried out by two independent authors (SE and PM) and conflicts were resolved by mutual consensus.

### 4.6. Study Quality and Risk of Bias

Study quality was ascertained by inclusion of studies using methodologies with standardised procedures, such as CLSI guidelines.

### 4.7. Data Extraction

Data were extracted from the included studies and organised in a table containing information on location of study, methodology used, objectives and relevant findings.

## 5. Conclusions

Thus, given that the majority of studies provide evidence in the form of MIC values, zones of inhibition diameters and time–kill studies, it can be concluded that *E. globulus* essential oil does possess antibacterial properties against MRSA and its biofilm. Despite many variations between the values published by the studies, there seems to be consensus that eucalyptus essential oil has bactericidal properties. Additionally, the studies also demonstrate that these antibacterial properties can be enhanced by combining the oil with other agents, such as other essential oils and antibiotics. Numerous compounds have been extracted from the plant, each with varying levels of antimicrobial activity, allowing for potency to be improved through isolation and combination of such constituents. These results provide evidence that supports the potential use of eucalyptus essential oil as an antimicrobial agent to combat the rapidly evolving problem of antimicrobial-resistant bacteria. More data are needed on this topic to draw conclusive results. Additionally, adopting a more standardised approach between studies would further enhance the reproducibility and accuracy of the results.

## Figures and Tables

**Figure 1 antibiotics-12-00474-f001:**
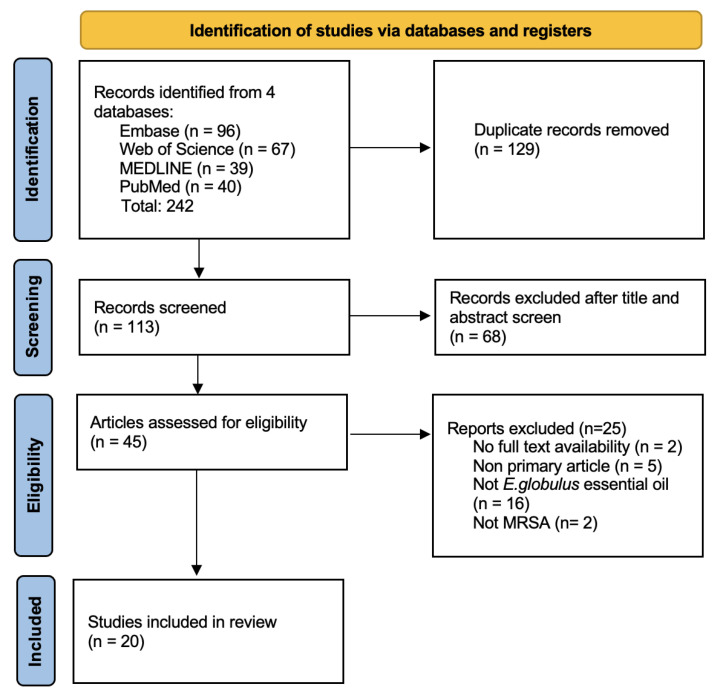
Preferred Reporting Items for Systematic Reviews and Meta-Analysis (PRISMA) flow diagram of study selection process.

**Table 1 antibiotics-12-00474-t001:** Antibacterial properties of *Eucalyptus globulus* essential oil against MRSA: a systematic review.

Author, Year, Location	Methodology	Objective	Intervention	Findings
Acs et al., 2016 [13]; Hungary	Chemical analyses of Eos—GC-FID and GC-MS Antimicrobial susceptibility—Disc diffusion In vitro tube dilution and vapour phase technique	To evaluate the antibacterial effect of various EOs against pathogens responsible for nosocomial respiratory tract infections	EOs: cinnamon bark, citronella, clove, eucalyptus, peppermint, Scots pine, thyme	GC-MS revealed eucalyptus oil contained 1,8-cineole (84.8%) Tube dilution—eucalyptus oil showed antibacterial effects in liquid medium with MIC—5.6 mg/mL and MBC—11.3 mg/mL Vapour phase: eucalyptus oil did not present inhibition against any test bacteria even in a 1500 μL/L concentration which differs from the tube dilution results
Ali et al. (2022) [14]; Pakistan	Antibiotic sensitivity testing—CLSI 2020 manual Antibacterial activity of EO—Well diffusion MIC	To identify the therapeutic potential of various plant EOs against MRSA	EOs: *Syzygium aromaticum*, *Eucalyptus globulus*, *Cinnamomum verum*, *Ferula assafoetida*	Eucalyptus oil recorded the lowest MIC, mean MIC 0.33 ± 0.11 mg/mL *p* > 0.05 Eucalyptus oil recorded second highest zone of inhibition: 18.67 ± 2.51 mm
Bouras et al. (2016) [15]; Algeria	Sensitivity test of EO and aqueous extract-Agar disc diffusion MIC and MBC—Agar dilution	To evaluate the antibacterial activity of EO and aqueous extract derived from *E. globulus* leaves	EO and aqueous extract of *E. globulus* leaves	The antibacterial activity of EO significant, but was lower than that of aqueous extract Disc diffusion EO: between 8 and 14 mm; very sensitive (++) for all strains. Highest activity of EO was observed against MRSA59 (12.00 ± 0.00 for 5 µL and 15.50 ± 0.70 for 10 µL). The best MIC and MBC of EO was 200 mg/mL
Chao et al. (2008) [16]; USA	Zone of inhibition—Disc diffusion assay in accordance with the Manual of Clinical Microbiology of the American Society for Microbiology	To screen EOs for inhibitory activity against MRSA to determine their potential for use as disinfectants, antiseptics or topical treatments	91 EOs alone, including *E. globulus* oil	Zone of inhibition eucalyptus oil—0 mm
Cui et al. (2021) [17]; UK	MIC as per CLSI Synergistic effects of oils and antibiotic—Modified Well Diffusion Time-kill assay	To determine various synergistic combinations for antimicrobial therapies as a potential strategy for treatment of multidrug resistant infection	29 plant EOs alone and EO—antibiotic combinations	MIC eucalyptus oil alone—0.313 *v*/*v*%, optimum concentration—1.56 *v*/*v*% Low level of synergy between eucalyptus oil and various antibiotics such as vancomycin, streptomycin, gentamicin, and tetracycline
Farsi and Alaidaroos, (2022) [18]; Saudi Arabia	Synthesis of silica nanoparticles (SiNPs) and oil encapsulation Evaluation of distribution of size, shape, and aggregation state SiNP—Transmission Electron Microscopes Determination of antimicrobial activity of eucalyptus EO with and without SiNPs—Agar well diffusion	To evaluate antibacterial efficacy using eucalyptus EO loaded on SiNPs against various pathogenic bacteria.	Eucalyptus oil alone, Silica dioxide nanoparticles alone and combination of both	Eucalyptus oil inhibition zones diameter: 11.33 ± 0.66 mm Silica dioxide nanoparticles alone: antibacterial activity appeared high with diameters of inhibition zones 14.33 ± 0.33 mm Eucalyptus oil loaded on silica dioxide nanoparticles revealed significant increase of the diameters of inhibition zones (*p* < 0.001) compared with eucalyptus EO only and SiNPs only −18.66 ± 0.33 mm
Hamoud et al. (2012) [19]; Germany	GLC-MS analysis MIC and MMC—determined by micro dilution method according to the German DIN regulation 58940-8 Time-kill assay	To investigate the antimicrobial activities of Olbas^®^ Tropfen (a complex EO distillate) in comparison to its isolated EO ingredients	Olbas (10 g) consists of peppermint oil (5.3 g), eucalyptus oil (2.1 g), cajuput oil (2.1 g) juniper berry oil (0.3 g) and wintergreen oil (0.2 g) + EOs individually	The main component of eucalyptus oil is 1,8 cineole (81.93%) Olbas showed significant antimicrobial activity against MRSA with MIC values of 0.15–20 mg/mL. In most cases, MMC values were one to two-times higher than MIC values, demonstrating a dose-dependent effect Eucalyptus oil: MIC: 10 mg/mL, MMC: 20 mg/mL Antimicrobial activity of Olbas and its EO ingredients can be ranked: juniper berry oil < wintergreen oil < eucalyptus oil < cajuput oil < Olbas < peppermint oil.
Hendry et al. (2012) [20]; UK	Establishment of microbial biofilms on stainless steel discs Determination of antimicrobial efficacy of the wipes—Agar diffusion assay Removal of microbial surface contamination by wipes and potential to promote cross-contamination	To investigate the antimicrobial efficacy of 5% eucalyptus oil and 2% eucalyptus oil containing wipes, specifically, its ability to remove microorganisms from hard surfaces, induce cross contamination and potential to eliminate bacterial biofilms	Wipes containing 5% and 2% EO, 2% CHG and 70% isopropyl alcohol (IPA)	No significant difference (*p* < 0.05) between the 5% and 2% eucalyptus oil formulations in their ability to remove microorganisms from steel surfaces, however both significantly (*p* < 0.05) removed more than the control (water control wipe) formulations. Microbial biofilms eliminated within 10 min (*p* < 0.05) when exposed to 2% eucalyptus oil formulation and within 5 min for 2% eucalyptus oil, control >30 min. Neither the 5% nor the 2% eucalyptus oil containing wipes induced cross-contamination onto successively touched surfaces, remnant microbial viability was not demonstrated.
Hendry et al. (2009) [21]; UK	MIC of aqueous CHG *E. globulus* EO and 1,8-cineole as per CLSI Chequerboard assays Biofilm: Chequerboard assay to assess the antimicrobial activity of CHG in combination with eucalyptus oil and 1,8-cineole against microorganisms in biofilm	To compare the antimicrobial efficacy of crude eucalyptus oil with its major constituent 1,8-cineole alone and in combination with CHG, against various pathogens when grown in planktonic and biofilm modes of growth	Aqueous CHG, eucalyptus oil and 1,8-cineole alone and CHG in combination with eucalyptus oil and 1,8-cineole	Antimicrobial activity was demonstrated by CHG, eucalyptus oil and 1,8cineole CHG was significantly more active against microorganisms in both planktonic and biofilm modes of growth (*p* < 0.05). Crude eucalyptus oil was significantly more efficacious against microorganisms grown in suspension compared with 1,8-cineole (*p* < 0.05). Synergistic activity was demonstrated between CHG and both eucalyptus oil and 1,8-cineole against suspensions of MRSA MIC for MRSA suspension: eucalyptus oil—2 mg/mL, 1,8 cineole—64 mg/mL MIC for MRSA biofilm: eucalyptus oil—512 mg/mL, 1,8 cineole >512 mg/mL
Horvath et al. (2011) [22]; Hungary	GLC/MS analysis—to determine antibacterial properties of oils and their components Direct bioautography assay—zone of inhibition	To chemically characterise the EOs of thyme, clove, eucalyptus, tea tree and cinnamon bark by TLC and identify the antibacterial activity of the oils and their main components against MRSA strains	EOs of thyme, clove, eucalyptus, tea tree and cinnamon bark and their isolated main compounds	Main component of eucalyptus oil was 1,8 cineole (84.2%) Antibacterial properties of eucalyptus oil were weaker than that of thyme, clove, and cinnamon bark, with mean zone of inhibition diameter for all MRSA strains being: 0 mm for 1 μL oil, 2.5 mm for 5 μL oil and 6.5 mm for 10 μL oil
Iseppi et al. (2021) [7]; Italy	Agar disk diffusion assay as per CLSI MIC FIC was determined Time-kill studies EO activity on mature biofilm—The effects of EOs, antibiotics, and the EO–EO and EO–antibiotic combinations on 24 h formed biofilm	To investigate if certain plant products can produce antibacterial effects against antibiotic-resistant pathogens, both alone and in combination with traditional antibiotics to which the bacterial strains were resistant.	Eucalyptus oil alone, eucalyptus oil in combination with other oils, eucalyptus oil in combination with oxacillin	Eucalyptus EO inhibitory zone diameter: 11–20 mm MIC: eucalyptus EO alone—showed good activity against 7 out 9 MRSA strains (MIC 0.032 mg/mL) FIC: eucalyptus EO—0.25, eucalyptus EO + Oxacillin—2.0, eucalyptus EO + Melaleuca alternifolia EO (TTO)—0.5 TTO and eucalyptus EO showed synergistic effect with the greatest reduction of biofilm in combination (*p* < 0.001) Time-kill studies also showed synergistic activity between TTO-eucalyptus EO and eucalyptus EO–Oxacillin showed the greatest synergistic effect, with bacterial load reduction obtained at low concentrations of both synthetic and natural compounds
Kwiatkowski et al. (2019) [23]; Poland	MIC as per CLSI Checkerboard method to identify synergy between 1,8-cineole and mupirocin	To investigate the antibacterial activity of EO compounds on mupirocin-susceptible and induced low-level mupirocin-resistant MRSA strains.	Isolated EO compounds (1,8-cineole, eugenol, carvacrol, linalool, (-)-menthone, linalyl acetate, and trans-anethole) in combination with mupirocin	Synergy between mupirocin and 1,8-cineole against both strains with FICI mupirocin sensitive strain—0.44, low level resistance against mupirocin strain—0.28 1,8-cineole and mupirocin combination decreased the MIC of 1,8-cineole from 307 ± 132 to 57.56 ± 0.00 mg/mL and from 57.56 ± 0.00 to 14.39 ± 0.00 mg/mL, respectively, for the MRSA^MupS^ and MRSA^MupRL^ strains
Merghni et al. (2018) [24]; Tunisia	Agar disk diffusion assay to determine antibacterial activity MIC and MBC—broth dilution method Inhibition of cell attachment—anti-adhesion properties tested following a microplate biofilm assay Antiquorum sensing	To identify antibacterial, antibiofilm and antiquorum sensing potential of *E. globulus* EO and its main component, 1,8-cineole against MRSA strains	*E.globulus* EO and its main component 1,8 cineole	*E. globulus* EO had a greater bacteriostatic effect than 1,8 cineole, However, EO had a reduced zone of inhibition compared to 1,8 cineole. EO zone of inhibition ranged from 10.7 ± 0.6 mm to 26.3 ± 0.6 mm 1,8-cineole ZOI ≥ 29 mm MIC: *E. globulus*—10 mg/mL, 1,8-cineole- 1.25 mg/mL Biofilm eradication—1,8-cineole was more effective against *S. aureus* 6538 and Sa18 than the EO Antiquorum sensing activity—EO was more effective than 1,8 cineole, even at low concentrations (MIC/4)
Mulyaningsih et al. (2011) [25]; Germany	The MIC of the samples was determined by broth microdilution methods The chemical composition of the fruit oil of *E. globulus* (EGF) was determined by GLC-MS, in comparison to the leaf oils from *E. globulus* (EGL)	To examine the antimicrobial activity of the fruit and leaf oil of *E. globulus*, *E. radiata* (ERL) and *E. citriodora* (ECL) against multidrug resistant bacteria. The major components of the oils were also isolated to identify a relationship between their chemical composition and antimicrobial properties.	*E. globulus* EO from fruits and leaves, individual components of the oil—aromadendrene, 1,8-cineole, citronellal, and citronellol	Main chemical compounds: Fruit oil—aromadendrene (31.17%) followed by 1,8-cineole (14.55%). Leaf oil—1,8-cineole (86.51%), α-pinene (4.74%) MIC: EGL- 2000 to >4000 µg/mL, EGF—0.25 and 1 mg/mL EGF exerted the most pronounced activity against methicillin-resistant *S. aureus* Among the four oils tested, the antimicrobial activity of the oils can be ranked as EGF > ECL > ERL > EGL
Mulyaningsih et al. (2010) [26]; Germany	GLC/MS analysis MIC/MBC—Broth microdilution assays as per CLSI Checkerboard method for synergistic, additive or antagonistic effects of combinations of individual compounds at different concentrations. Fractional inhibitory concentration indexes Time-kill experiments as per results of the checkerboard assay	To investigate *E. globulus* fruit (EGF) oil and its three major components (aromadendrene, 1,8-cineole, and globulol) against antibiotic-susceptible and antibiotic-resistant microorganisms and test their synergistic effects when applied in combination	EGF alone, the main constituents alone (aromadendrene, 1,8-cineole, and globulol) and main constituents in combinations at different concentrations	Aromadendrene was the most abundant compound of EGF (31.17%) followed by 1,8-cineole (14.55%), globulol (10.69%), and ledene (7.13%) EGF oil showed the most potent antibacterial activity with MIC of 0.12–1 mg/mL, followed by aromadendrene with 0.25–1 mg/mL Aromadendrene showed higher antimicrobial properties than 1,8-cineole and globulol (*p* < 0.05 in both cases). Gobulol and 1,8 cineole showed low antimicrobial activities Synergy was noted at 0.12 mg/mL aromadendrene plus 16 mg/mL 1,8-cineole for MRSA
Punitha et al. (2014) [27]; India	Antibiotic susceptibility test—Disc Diffusion method of Kirby Bauer on Muller-Hinton agar as per CLSI Biofilm production assay—Congo red agar method Antibacterial screening—agar well diffusion method	To evaluate whether essential oils could inhibit the growth of *S. aureus* biofilm forming isolates	EOs (Eucalyptus, Mint, Turpentine, Neem and Amla)	Inhibition against biofilm-forming MRSA isolate’s—average zones of inhibition ranged from 12.2 ± 1.77 mm to 26.2 ± 1.93 mm. Multiple isolates were inhibited considerably by eucalyptus oil. Turpentine oil was most active against biofilm-forming MRSA followed by eucalyptus, mint, and neem
Salem et al. (2018) [28]; Tunisia	Gas chromatography analysis Antibacterial activity—disk diffusion method MIC—microdilution method Synergistic interaction by determining fractional inhibitory concentration index (FICI)—2D checkerboard method	To evaluate the antioxidant, antimicrobial, and cytotoxicity properties of *E. globulus* EOs and assess synergy between the EOs and conventional antimicrobials	*E.globulus* EOs at vegetative, full flowering and fructifications stages + synergy between *E. globulus* EOs and conventional antimicrobials	67 volatile compounds were identified as part of the oil. The chemical composition differed depending on the plant stage. 1,8-cineole was the major compound at vegetative and full flowering (32.19%), *p*- cymene was the major compound at fructification stages (37.82%) Each stage also had a different level of antibacterial activity, with the vegetative stage possessing the most potent antibacterial properties. MIC: vegetative 2 mg/mL, full flowering 4 ± 1.3 mg/mL, fruitification 4 ± 0.0 mg/mL Inhibition zones reached up to 38 mm during vegetative stage, 24 mm during flowering and fruitification. The combination of eucalyptus EO with ampicillin revealed a partial synergistic effect against MRSA with FICI of 0.53
Simsek and Duman, (2017) [29]; Turkey	MIC determined FIC determined	To compare the antimicrobial efficacy of 1,8-cineole, alone and in combination with CHG against various microorganisms	1,8-cineole isolated from *E. globulus* alone and in combination CHG	CHG combined with 1,8-cineole showed synergistic antimicrobial activity in some of the tested microorganisms. The MIC values of CHG alone and in combination with 1,8-cineole were found as, 0.004 mg/mL and 0.000125 mg/ML for MRSA MIC values of 1,8-cineole alone and in combination with CHG were 128 mg/mL and 8 mg/mL for MRSA
Tohidpour et al. (2010) [30]; Iran	Antibacterial susceptibility—disc diffusion GC-MS—chemical composition MIC—agar dilution method as approved by NCCLS with minor modification	To test the antibacterial effect of EOs from *T. vulgaris* and *E. globulus* against MRSA and analyse the biochemical composition of the oils	Eucalyptus oil EO and *Thymus vulgaris* EO alone	Inhibition zones for *E. globulus*: 8 mm, Vancomycin: 15 mm, Methicillin: 0 mm, cotrimoxazole: 0 mm Mean MIC for *E. globulus* 0.0856 mg/mL Eucalyptol (47.2%), (+) Spathulenol (18.1%) and a-Pinene (9.6%) were the major compounds of the eucalyptus EO.
Warnke et al. (2009) [31]; Germany, Australia, UK	Antibacterial activity—disk diffusion method	To identify the antibacterial efficacy of various essential oils on frequently isolated and hospital-acquired MRSA	13 EOs alone	Inhibition zone *E. globulus*—14 mm

Abbreviations: GC-FID—gas chromatography flame ionization detector, GC-MS—gas chromatography mass spectrometry), EO—essential oil, CLSI—*Clinical and Laboratory Standards Institute*, MIC—minimum inhibitory concentration, MMC—minimum microbicidal concentration, MBC—minimum bactericidal concentration, MRSA—methicillin resistant *Staphylococcus aureus*, FIC—fractional inhibitory concentration, FICI—fractional inhibitory concentration index, CHG—chlorhexidine digluconate. NCCLS—National Committee for Clinical Laboratory Standards.

## Data Availability

All data are contained within this article.

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
