# Peer review of "Antibacterial Properties of Eucalyptus globulus Essential Oil against MRSA: A Systematic Review"

_antibiotics, 2023, doi:10.3390/antibiotics12030474_

Round 1
Reviewer 1 Report
The manuscript entitled "Antibacterial properties of Eucalyptus globulus essential oil against MRSA: A systematic review." is written as per the style. There are a few corrections and queries related to the unit's like mg/ml or mcg/ml; nnm or nm. These can be corrected accordingly,
The plagiarism is 20%, however it was noted that the manuscript was submitted to the Turnitin.com website through university.
The minor corrections are mentioned in the reviewed pdf file which can be corrected and updated.

Author Response
We thank the reviewers for their insightful comments and believe these have substantially improved the manuscript. Please find our point-wise response below:
- The manuscript entitled "Antibacterial properties of Eucalyptus globulus essential oil against MRSA: A systematic review." is written as per the style. There are a few corrections and queries related to the unit's like mg/ml or mcg/ml; nnm or nm. These can be corrected accordingly,
Response: Thank you for your comment. We have corrected these units in the manuscript.
- The plagiarism is 20%, however it was noted that the manuscript was submitted to the Turnitin.com website through university.
Response: Thank you for your comment. Efforts have been made to have the similarity percentage as small as possible and we have supported our text extensively with citations. We believe journals perform their own plagiarism checks and when needed they notify authors to make changes to reduce similarity percentage. Furthermore, journals also look at the manuscript in its entirety to make such decision. We have not received any instructions from the journal to reduce similarity percentage.
- The minor corrections are mentioned in the reviewed pdf file which can be corrected and updated.
Response: These have been corrected and updated i.e. units for Bouras et al; Zone of inhibition for Chao et al; MIC values for Kwiatkowski et al; units for Mulyaningsih et al; units for Simsek and Duman; and nm in section 3.2.6.
Reviewer 2 Report
The review entitled "Antibacterial properties of Eucalyptus globulus essential oil against MRSA: A systematic review" is presented for the peer review.
In the study, authors highlighted antibacterial properties of Eucalyptus globulus essential oil against MRSA. Authors provides some of various literatures to investigate the antibacterial properties of E. globulus essential oil either alone or in combination with other agents against MRSA.
The comments are given below.
Line 33: The abbreviation "WHO" should be used immediately after "World Health Organization".
Line 131: Should the abbreviation "MIC" be "minimum inhibitory concentration" instead of "mean inhibitory concentration"?
Line 139: Should be more appropriate to give the reference numbers in the text in single parenthesis instead of (13), (22), (19), (25, 26), (30).
Line 154,155 References must be numbered in order of appearance in the text and listed individually at the end of the manuscript. Also in the text, reference numbers should be placed in square brackets [ ], and placed before the punctuation.
Although the topic of antimicrobial resistance and the need for alternative agents to combat the problem is of great importance in public health, the methodology used in the systematic review has some limitations that may affect the reliability and generalizability of the results. The use of various methods across twenty studies, although providing some evidence, may not be sufficient to draw definitive conclusions on the effectiveness of E. globulus essential oil against MRSA. Furthermore, the limited number of studies included in the review may not fully capture the current state of research in this field. To improve the quality of the review, the authors may consider expressing this at the end of the discussion.
The review provides valuable information on a relevant topic and can be considered for publication after undergoing this revisions.
Author Response
We thank the reviewers for their insightful comments and believe these have substantially improved the manuscript. Please find our point-wise response below:
The review entitled "Antibacterial properties of Eucalyptus globulus essential oil against MRSA: A systematic review" is presented for the peer review. In the study, authors highlighted antibacterial properties of Eucalyptus globulus essential oil against MRSA. Authors provides some of various literatures to investigate the antibacterial properties of E. globulus essential oil either alone or in combination with other agents against MRSA.
The comments are given below.
- Line 33: The abbreviation "WHO" should be used immediately after "World Health Organization".
Response: Thank you for the suggestion. We have edited it and put WHO immediately after World Health Organisation. The text now reads as “It is a major cause of global mortality and is classified by the World Health Organisation (WHO) as being in the top ten threats to global health.”
- Line 131: Should the abbreviation "MIC" be "minimum inhibitory concentration" instead of "mean inhibitory concentration"?
Response: Thank you for pointing this out. The abbreviation ‘MIC’ is indeed ‘minimum inhibitory concentration’. Corrections have been made under abbreviations of Table 1 by replacing ‘mean’ with ‘minimum’. It is also corrected in the first line of section 4.2.
- Line 139: Should be more appropriate to give the reference numbers in the text in single parenthesis instead of (13), (22), (19), (25, 26), (30).
Response: Thank you for your suggestion. We have put these reference numbers in a single parenthesis.
- Line 154,155 References must be numbered in order of appearance in the text and listed individually at the end of the manuscript. Also in the text, reference numbers should be placed in square brackets [ ], and placed before the punctuation.
Response: Thank you for your comment. We have tried to list references in order of appearance and list them with individual numbers in the bibliography. Table 1 precedes most of the Results section and many references cited in the results section have already appeared in Table 1. However, we have corrected the reference numbers by placing them before punctuations as indicated by track changes. We have inserted references by EndNote which puts them in round brackets. In case this manuscript is accepted and there is a strict requirement for square brackets, we believe the editing team will have a system in place to convert round brackets to square brackets. If not, we will remove EndNote and manually convert them to square brackets.
- Although the topic of antimicrobial resistance and the need for alternative agents to combat the problem is of great importance in public health, the methodology used in the systematic review has some limitations that may affect the reliability and generalizability of the results. The use of various methods across twenty studies, although providing some evidence, may not be sufficient to draw definitive conclusions on the effectiveness of E. globulus essential oil against MRSA. Furthermore, the limited number of studies included in the review may not fully capture the current state of research in this field. To improve the quality of the review, the authors may consider expressing this at the end of the discussion.
Response: Thank you for your suggestion. To address it, we have added new text at the end of the Discussion which reads as “Furthermore, there is limited research on this topic and thus the number of studies included in this review is minimal. This diminishes the ability to make conclusive judgments and thus more data is required. Within these studies, there are a wide variety of methods used, which although providing valuable insight into various paraments, affects the overall reliability and applicability of the data.”
- The review provides valuable information on a relevant topic and can be considered for publication after undergoing this revisions.
Response: Thank you for your feedback. We have tried our best to revise the manuscript according to reviewers suggestions.
Reviewer 3 Report
- The introduction part needs improvement regarding essential oils as antibiotics.
- Why did the author choose Eucalyptus globulus to conduct the study?
- Parameters considered for the study need to be described.
- Only the combination of EO criteria is considered by the author.But, what about the combination of EO with antibiotics? In the abstract and conclusion, indicate your conclusion.
- In the line 55–57The mechanism of action of essential oils differs from the mechanism of antibiotics in that it inhibits various Mention the main antimicrobial routes for essential oils.
- How is oil described better than antibiotics?
- Many mistakes exist, such as AND (eucalyptus oil OR eucalyptus OR aromaden-87 drene OR 1,8 cineole OR globulol) AND----
- The punctuation is poorly used in the manuscript, so I suggest improving the level of language in the manuscript.
- Use scientific pattern in the scientic name of organism-like in the line 89-resistant s aureus, in the line-100--that tested Eucalyptus species other than--this need to check through out the manuscript
- Figure one is not clear as in the body of manuscript ===described exclusion of review articles but in figure it is mentaion as included?
- In the table-1 in the section MIC—insert the name of pathogen as MIC was recorded.
- What are the reasons that eucalyptus leaf oil has much higher activity as aromadendrene-but aromadendrene is less potent, described in paragraph-Additionally, the MIC of eucalyptus leaf oil was 16 times higher 323 than that of fruit oil. The fruit oil had the same MIC as aromadendrene-------please make more clear way
- Conclusion part is need to described in more significant way.
Author Response
We thank the reviewers for their insightful comments and believe these have substantially improved the manuscript. Please find our point-wise response below:
- The introduction part needs improvement regarding essential oils as antibiotics.
Response: Thank you for your suggestion. To address it, we have added text in the Introduction section which reads as “Examples of targets include inhibition of cell membranes, efflux pumps, biofilm and motility. The combination of various mechanisms complements one another and lead to greater inhibition of bacterial growth compared to traditional antibiotics.”
- Why did the author choose Eucalyptus globulus to conduct the study?
Response: Thank you for your comment. As indicated in the last paragraph of the Introduction section, we chose Eucalyptus globulus because the essential oil derived from it is widely used in various industries including pharmaceuticals, perfumes, food products and cosmetics. Though its antibacterial properties against a range of bacteria are known but its effectiveness against MRSA is not known and this is what we wanted to explore in this study.
- Parameters considered for the study need to be described.
Response: Thank you for your comment but we are not sure what is meant by parameters here. If this pertains to eligibility criteria, we have section 2.4 in the Methods where we have defined inclusion and exclusion criteria of the study. Furthermore, the focused question, PICO question and search strategy of the study is clearly defined in the Methods section.
- Only the combination of EO criteria is considered by the author.But, what about the combination of EO with antibiotics? In the abstract and conclusion, indicate your conclusion.
Response: Thank you for your comment. To address it, we have edited the text in the conclusion section which now reads as “Additionally, the studies also demonstrate that these antibacterial properties can be enhanced by combining the oil with other agents such as other essential oils and antibiotics.”
We already have text to this effect in the abstract section which reads as “The findings suggest that E. globulus essential oil has antibacterial properties against MRSA which can be enhanced when used in combination with other agents such as other essential oils and antibiotics.”
- In the line 55–57The mechanism of action of essential oils differs from the mechanism of antibiotics in that it inhibits various Mention the main antimicrobial routes for essential oils.
Response: Thank you for this suggestion. To address it, the new added reads as “Examples of targets include inhibition of cell membranes, efflux pumps, biofilm and motility. The combination of various mechanisms complements one another and lead to greater inhibition of bacterial growth compared to traditional antibiotics.”
- How is oil described better than antibiotics?
Response: Sorry, we did not understand this comment because our manuscript does not describe eucalyptus essential oil to be better than antibiotics.
- Many mistakes exist, such as AND (eucalyptus oil OR eucalyptus OR aromaden-87 drene OR 1,8 cineole OR globulol) AND----
Response: These are the keywords for the search strategy. To our understanding, authors are supposed to list the keyword search string in a systematic review article hence we have given it the way it was used in literature search across various databases.
- The punctuation is poorly used in the manuscript, so I suggest improving the level of language in the manuscript.
Response: Thank you for pointing this out. We have corrected it at various places with track changes.
- Use scientific pattern in the scientic name of organism-like in the line 89-resistant s aureus, in the line-100--that tested Eucalyptus species other than--this need to check through out the manuscript
Response: In line 89 - these are the keywords for search strategy where variations of the name of the organism are included. It is notable that capitalizing the first letter of the genus as per scientific pattern does not affect the search results because key words can be written in capital or small letters. In the manuscript text, names of plants or organisms are spelt as per scientific norms. For the ‘Eucalyptus species’ in line 100, we have corrected it to ‘species of Eucalyptus’. We have also checked the rest of the manuscript for proper scientific names.
- Figure one is not clear as in the body of manuscript ===described exclusion of review articles but in figure it is mentaion as included?
Response: We believe the reviewer’s reference to the body of manuscript is directed towards section 2.4.2 where it is stated that review articles were excluded. This is true because review articles were not included in this study. The bottom box of Figure 1 shows ’studies included in review’ – these are the final 20 studies included in this systematic review.
- In the table-1 in the section MIC—insert the name of pathogen as MIC was recorded.
Response: Given the aim of this study, the only pathogen included is MRSA, so MICs given in Table 1 are for MRSA. This is clearly mentioned in the title of the table.
- What are the reasons that eucalyptus leaf oil has much higher activity as aromadendrene-but aromadendrene is less potent, described in paragraph-Additionally, the MIC of eucalyptus leaf oil was 16 times higher than that of fruit oil. The fruit oil had the same MIC as aromadendrene-------please make more clear way
Response: This point has been made clear by editing the end of second paragraph in section 4.2. The edited text now reads as “Additionally, the MIC of eucalyptus leaf oil was 16 times higher than that of fruit oil. As demonstrated by Mulyaningsih et al. 2011 (25) this may be explained by the fact that the main component of the leaf oil is 1,8 cineole, whereas the main component of the fruit oil is aromadendrene. Aromadendrene has more antibacterial potency than 1,8 cineole, resulting in a lower MIC of the leaf oil (25).”
- Conclusion part is need to described in more significant way.
Response: Thank you for your comment. To address it, we have added new text in the Conclusion section that reads as “Additionally, the studies also demonstrate that these antibacterial properties can be enhanced by combining the oil with other agents such as other essential oils and antibiotics. Numerous compounds have been extracted from the plant, each with varying levels of antimicrobial activity, allowing for potency to be improved through isolation and combination of such constituents. These results provide evidence that supports the potential use of Eucalyptus essential oil as an antimicrobial agent to combat the rapidly evolving problem of antimicrobial resistant bacteria. More data is needed on this topic to draw conclusive results. Additionally, adopting a more standardised approach between studies would further enhance the reproducibility and accuracy of the results”
Round 2
Reviewer 3 Report
Authors incorporated all raised comments and suggestion.